# Association between Social and Emotional Competencies and Quality of Life in the Context of War, Pandemic and Climate Change

**DOI:** 10.3390/bs13030249

**Published:** 2023-03-12

**Authors:** Fátima Gameiro, Paula Ferreira, Miguel Faria

**Affiliations:** 1Research Center in Social Work and Social Intervention (CISIS), Institute of Social Work, Lusófona University—University Center of Lisbon, Campo Grande, 376, 1749-024 Lisbon, Portugal; 2Centro de Investigação em Serviço Social e Intervenção Social, Institute of Social Work, Lusófona University—University Center of Lisbon, Campo Grande, 376, 1749-024 Lisbon, Portugal; 3Escola Superior de Saúde Ribeiro Sanches, Lusófona Polytechnic Institute, Rua do Telhal aos Olivais n8-8A, 1950-396 Lisbon, Portugal

**Keywords:** social and emotional competencies, quality of life, environment, canonical correlation

## Abstract

The present context, with an ongoing pandemic situation, war and climate change, seems to play a critical role in both the peoples’ perception of their quality of life, and the acquisition and development of social and emotional competencies. In this study, our goal was to assess the relationship between social and emotional competencies and peoples’ quality of life in a Portuguese sample. Participants were 1139 individuals living in Portugal, aged between 16 and 85 years old, who were mostly (73%) female. An online protocol for data acquisition was used, which included sociodemographic characterization, the Portuguese version of the scale of Social and Emotional Competencies (SEC-Q) and the World Health Organization Quality of Life (WHOQOL-BRIEF). Correlation analysis and a canonical correlation were performed, with results showing a high association between the dimensions of social and emotional competencies and peoples’ quality of life. Two significant canonical roots were extracted, and the results show that the first is characterized by internal factors, linking psychological health and self-management and motivation, and the second root evidences the external factors, linking social relations and environment with social awareness and pro-social behavior.

## 1. Introduction

Throughout the life cycle, the individual develops the skills, attitudes and values necessary to acquire social and emotional competencies (SECs) [1]. Defined as the ability to understand, manage, and express social and emotional issues, SECs include skills such as self-awareness (capacity to pay attention and understand one’s own emotions, goals and values, and being able to recognize the relationship between thoughts, feelings and behaviors), self-management (ability to manage one’s own emotions and behaviors to facilitate motivation and achieve one’s goals), social awareness (capacity to understand other people, different social contexts and norms), relationship skills (making it possible to initiate and maintain prosocial interpersonal relationships, respect social norms and have good communication skills), and responsible decision-making (reflexive consideration of different choices, taking into account the wellbeing of the self and others) [2,3]. These skills are considered essential to achieve the successful management of daily life [4] and a good quality of life [5].

Emotional competencies (EC) are related to mental health, academic performance, and/or the work context. In a successful development process, adaptive resilience emerges; it enables one to effectively overcome stressful life circumstances [6], improve psychological, somatic, and adjustment issues, decrease stress levels and somatic complaints, and increase the quality of social relationships [7]. Social competencies (SC) facilitate adaptive functioning, positive adjustment and the subsequent achievement of goals, despite the warnings or stress that may arise throughout life. These skills allow an increase in pro-social behaviors, facilitating healthy interpersonal relationships, decreasing delinquency, and promoting the individual’s well-being and mental health [8]. With the mastery of SECs, the individual is no longer predominantly controlled by external factors (external locus of control), and increasingly begins to act according to internalized beliefs and values (internal locus of control), to care and worry about others, to make good decisions, and to take responsibility for their choices and behaviors [9]. Thus, these skills enable the individual to understand and manage emotions and social interactions, decrease risky behaviors, and consequently promote personal health and well-being [3,10].

However, the environment and the conditions we live in influence our lifestyle, and have an impact on the perception of one’s quality of life (QoL). The World Health Organization (WHO) defines QoL as an individual’s perception of their position in life in the context of the culture and value systems in which they live and in relation to their goals, expectations, standards and concerns [11]. QoL comprises and is influenced by a physical domain (pain and discomfort, energy and fatigue, sleep and rest), a psychological domain (positive feelings, thinking, learning, memory and concentration, self-esteem, body image and appearance, negative feelings), one’s level of independence (mobility, activities of daily living, dependence on medication or treatments, working capacity), a social relationships domain (personal relationships, social support, sexual activity) and an environment domain (physical safety and security, home environment, financial resources, availability and quality of health and social care, opportunities for acquiring new information and skills, participation and opportunities for recreation and leisure, pollution, noise, traffic, climate, transport) [12].

The current context of pandemics, war and climate change seems to play a key role in the acquisition and development of SECs [13], and the perception of QoL is associated with high levels of stress and disaster situations. The literature has associated these last atypical years with significant damage to QoL [13,14,15,16,17,18,19,20,21].

COVID-19 has been identified as a traumatic event, as it created sudden changes in lifestyle and interpersonal relationships, economic problems and uncertainty about the future [13,22]. According to the literature, social isolation has affected various aspects of QoL [23], is positively correlated with high levels of distress [24], and has negatively impacted the construction and development of SECs [13]. During the pandemic, many individuals began to manifest symptoms of anxiety and depression, in addition to insomnia, stress, and changes in biological rhythm [25,26]. It is estimated that 40% to 50% of adults experienced psychological distress following the COVID-19 outbreak, and that 30% of adults and children are at risk of post-traumatic stress [27]. In another study [28], individuals reported moderate levels of physical QoL and low levels of QoL in terms of their social relations and their relationship with the environment caused by the lockdown measures decreed in Portugal. It should be noted that during the state of emergency, with some exceptions, Portuguese people were only allowed to travel to work, to acquire essential goods, to assist vulnerable people and to use health services [29]. Thus, the Portuguese population experienced moments of high social isolation [30,31], which was also associated with increased stress and difficulty in short/medium-term planning [32]. As a result, after the first lockdown, there was an increase in alcohol and cigarette consumption, reduced physical activity, and an increased consumption of unhealthy foods, factors that may have contributed to a decrease in subjective well-being [33]; indeed, these will have effects that will last over time, whether socially, economically, behaviorally and/or psychologically [34].

Some studies have focused on the QoL of civilian victims in the context of war, armed conflict and violence. Steel et al. [35] conducted a meta-analysis in which they explored 161 articles and reported rates of 30.6% and 30.8% in terms of the levels of symptoms associated with post-traumatic stress disorder and depression, respectively. They also found that different factors influence the QoL of victims of an armed conflict, including sociodemographic, social, and psychological characteristics. In 2019, in Colombia, Simancas-Fernández et al. [36] assessed QoL in the victims of armed conflicts and identified that the environmental domain was the most affected due to disadvantages related to physical, health, and social care resources. Campo-Arias et al. [37] conducted a systematic review of 13 studies that assessed the mental health of victims of armed conflicts and found that the prevalence of mental disorders was between 1.5 and 32.9%.

The United Nations Environment Program (UNEP) of the UN [38] conducted, in partnership with Oxford University, research in 50 countries and surveyed over 500,000 people. They found that about 64% of the respondents believed climate change to be a global emergency. According to Figueiredo [39], global health, climate and ecological conditions are inextricably linked, and that over the past decade, the health impacts of climate change have been deeply felt. Thus, war and the perception of the short and medium-term consequences of climate change are associated with stressful events, and seem to be related to SECs and QoL. In a study of 105 farmers in Switzerland, in 2019, regarding climate change, the majority revealed that they perceive difficulties related to self-efficacy, self-control, and social skills [40].

Given the current world scenario, where pandemics, war and climate change are a reality, and with the analysis of the literature showing an interaction between SECs and QoL, we intended to assess the relationship between the perception of social and emotional competencies and the perception of QoL in a sample of Portuguese individuals.

For this, a quantitative study was carried out, whose methodological procedure and results are presented and discussed below.

## 2. Materials and Methods

### 2.1. Participants

This investigation included 1139 participants who were living in Portugal and were aged between 16 and 85 years old. Two inclusion criteria were defined: being 16 years of age or older (according to the Portuguese legislation about informed consent, which waives guardians or legal representatives’ authorization to participate in the study) and having resided in Portugal for more than a year (according to permanent resident status).

The majority were Portuguese (93.9%), female (73.3%), aged between 18 and 39 (48.5%), single (50.6%), with a degree in higher education (44.8%), employed (60.7%) and living in a town (65.4%) (see Table 1).

### 2.2. Instruments

A sociodemographic questionnaire that includes gender, age, nationality, marital status, educational level, professional occupation and locale of residence was used.

Social and emotional competencies were assessed using the Social and Emotional Competencies Questionnaire (SEC-Q), which was developed by Zych et al. [3] and adapted to the Portuguese population by Lobo [41]. It seeks to assess social and emotional competencies from the individual’s own perspective (self-report), taking the last 12 months as a reference. It consists of 16 items divided into four components: self-awareness, self-management and motivation, social awareness and pro-social behavior, and decision-making. SECs were evaluated with a 5-point Likert scale ranging from 1 (strongly disagree) to 5 (strongly agree). Zych et al. [3] found that the SEC-Q presented good psychometric qualities in two samples, one with 643 university students (α = 0.87) and another with 2139 adolescents (α = 0.80). In our study, the Cronbach alpha was 0.87.

To evaluate QoL, the World Health Organization Quality of Life–Bref (WHOQL-BREF) was used. Originally developed by the World Health Organization Quality of Life Group in 1994 [42], we used the Portuguese version, adapted by Canavarro et al. [43]. It consists of four domains: physical, psychological, social relationships and environment, with 26 items evaluated using a five-point Likert scale: for questions 1 and 15: from 1—“Very poor” to 5—“Very good”; for questions 2, 16 to 25: from 1—“Very dissatisfied” to 5—“Very satisfied”; for questions 3 to 9: from 1—“Not at all” to 5—“Very much”; for questions 10 to 14: from 1—“Not at all” to 5—“Completely”; and for question 26: from 1—“Never” to 5—“Always” [43]. The Portuguese version of this instrument presents a good internal consistency, with a Cronbach’s alpha value of 0.92 [43]. In our study, that value was 0.89.

### 2.3. Procedure

After institutional approval, the study was made available on social networks. After informed consent was obtained, the protocol with the three questionnaires mentioned above was answered. This procedure took place between July and September 2022. From this application, 1140 questionnaires were received, one of which was excluded for being incomplete, leaving us with 1139 valid cases.

### 2.4. Statistical Analysis

The statistical analyses were performed with the Statistical Package for the Social Sciences (IBM, SPSS Statistics, version 28.0 for Windows). First, we found the descriptive values for both instruments; then, the correlations between the dimensions of the WHOQOL_BREF and the SEC_Q were calculated. Lastly, with the aim of understanding the relations between the dimensions of the WHOQOL_BREF and the SEC_Q, and the contribution of these dimensions, a canonical correlation was performed.

## 3. Results

### 3.1. Descriptive Values

The perception of social and emotional competencies as a whole was at a medium–high level (M = 3.92; SD = 0.45), with self-awareness (M = 3.98; SD = 0.54), social awareness and pro-social behavior (M = 3.97; SD = 0.50) being the dimensions with the highest mean scores (see Table 2).

We can also see that the perception of quality of life as a whole in our sample was at an average level (M = 3.75; SD = 0.48). However, in its individual dimensions, the highest value refers to the physical domain (M = 3.89; SD = 0.59), followed by social relations (M = 3.79; SD = 0.71) and the psychological domain (M = 3.70; SD = 0.64); meanwhile, the environment was the domain with the lowest values (M = 3.63; SD = 0.53) (See Table 2).

### 3.2. Correlations

The four dimensions of the WHOQOL presented positive and moderate correlations among themselves, with values of Pearson´s r ranging from 0.380 (physical health and social relationships) to 0.547 (physical health and psychological health). All four dimensions correlated strongly with the WHOQOL total score, as expected, with values above 0.730.

A similar pattern of correlations was found among the four dimensions of the SEC_Q, with Pearson´s r values between 0.345 (self-awareness and decision-making) and 0.478 (self-management and motivation and social awareness and pro-social behavior). Again, all these dimensions were strongly correlated with the SEC_Q total score, with values between 0.698 (self-awareness) and 0.836 (social awareness and prosocial behavior), as shown in Table 3.

As for the correlations between the several dimensions of the WHOQOL and the SEC_Q, we can see that they were all positive and weak to moderate, with values ranging from 0.250 (environmental health and decision-making) to 0.577 (psychological health and self-management and motivation). All these correlations were significant at the 0.01 level.

These results suggest that quality of life and social and emotional competencies are strongly related. However, with the exception of the correlation between psychological health and self-management and motivation (r = 0.577), all other values of Pearson´s r were very similar, making it difficult to know which specific dimensions of quality of life are related to the specific dimensions of the social and emotional competencies. To investigate these relations, we performed a canonical correlation.

### 3.3. Canonical Correlation

A canonical correlation analysis was conducted using the four dimensions of the social and emotional competencies as predictors of the four dimensions of quality of life in order to evaluate the multivariate shared relationship between the two variable sets (emotional and social competencies and quality of life). Four functions were obtained, with canonical correlations of 0.619, 0.263, 0.063 and 0.007 for each successive function. The full model across all functions was statistically significant using the Wilks’s λ = 0.572 criterion, F(16, 3419.239) = 42.870, *p* < 0.001. Since the value of the Wilks statistic represents the variance unexplained by the model, we can see that the value of the effect size in an r2 metric is 0.428 (1 − λ), indicating that this model explains almost 43% of the variance shared between the variable sets.

Given the explained variance for each function, only the first two functions were considered noteworthy in the analysis (38.3% and 6.9% of shared variance, respectively). The last two functions together only explained less than 1% and were therefore discarded (see Table 4).

The standardized canonical function coefficients and structure coefficients (canonical loadings) for Functions 1 and 2 are shown in Table 5. The squared structure coefficients are also given, as well as the communalities (h2) across the two functions for each variable.

Looking at the coefficients of Function 1, we can see that psychological health is the relevant variable in the criterion, what is supported by the analysis of the structure coefficients (canonical loadings). In a secondary position, we can see the remaining three dimensions of quality of life (physical, social relations and environment). The fact that they all share the same sign indicates that these variables are all positively correlated (see Table 5).

Regarding the predictor variable set in Function 1, the variable that stands out is self-management and motivation, with all the others presenting lower coefficients, also with negative signs.

This result is not surprising, since the analysis of the correlations matrix had already indicated that the strongest association between the dimensions of the two instruments occurred between psychological health and self-management and motivation; this indicates what might be described as a positive state of mind, including both the psychological health and the drive to achieve significant goals, within reasonable boundaries, corresponding to self-management and motivation.

In Function 2, we can see that the criterion variables that contribute the most are social relations and environment (confirmed by the analysis of the structure coefficients), while in the predictor variable set, the variable to consider is social awareness and pro-social behavior. This second function clearly enhances a different perspective when compared with the previous. While in the first function we had what might be described as an internal attitude, now we can see some factors that clearly point to the external dimension of the individuals, through their behaviors and social relationships.

In both variable sets, it is visible that psychological quality of life and motivation, which were the more important variables in Function 1, still present high coefficients, but that their canonical loadings are weak, which happens probably because of some degree of collinearity [44]; therefore, their contribution to the canonical variates should not be considered (see Table 5).

## 4. Discussion

In the current context, the world population, and more specifically the Portuguese population, is under a high level of stress and, according to some authors, experiencing situations considered as catastrophic,; these negatively affect the perception of one´s quality of life, as well as one’s social and emotional competencies. In view of this reality, this study aimed to assess the relationship between the perception of personal and social competencies and the perception of quality of life in a sample of Portuguese individuals.

The context in which the individual lives seems to play a key role in the acquisition and development of SECs [45]. According to Soto-Sanz et al. [46], individuals with more SECs are more protected against the impact of disasters. Based on this premise, it was possible to verify in a sample of Portuguese individuals that they show a median perception of social and emotional competencies; they are high in self-awareness, which is followed by social awareness and pro-social behavior, self-management and motivation, and they are low in the decision-making domain. According to the literature, social isolation affected several aspects of individuals’ quality of life, and is often associated with high levels of distress, having a negative effect on the construction and development of SECs [13]. In this study, individuals perceived self-awareness as the highest competence and the decision-making domain as the lowest. The way the pandemic was managed by national entities and conveyed by the media was appealing to individuals’ social awareness, and to their individual and collective responsibility. However, it seems that these guidelines were mostly translated into a need for the subject to think about and feel the pandemic, focusing more on the problem and not on the resolution; this seems to have generated a greater level of difficulty in terms of making decisions. This reality resulted in increased symptoms of anxiety and depression, in addition to insomnia, stress and changes in biological rhythm [25,26].

In terms of quality of life, this study confirmed that a lower perception of quality of life was found in the environmental domain. These results are in line with the literature, which has shown the impact that stressful or traumatic situations (as was the case of the pandemic) have on the QoL of individuals, especially in terms of social relations and their relationship with the environment [28]. Borine et al. [47] and Tenais and Ribeiro [29] found similar results, caused by the lockdown measures enacted in Portugal. In addition, Simancas-Fernández et al. [36], when assessing victims of the armed conflict, identified that the environmental domain was the most affected.

In the Portuguese sample, the predictors of the relationship between SECs and QoL are mostly associated with factors that exist within the internal locus of control (explaining approximately 38% of this relationship), specifically, psychological health (in QoL) and self-management and motivation (in SECs). Thus, the ability to manage one’s emotions and behaviors in order to facilitate motivation and achieve one’s goals is fundamental to creating positive feelings, enhancing thinking, learning, memory and concentration, self-esteem, body image and appearance, and adequately managing negative feelings. In this period of instability, impositions are a constant reality, routines are changed and decisions are conditioned [29], and the self-management capacity of individuals and their own motivation is changing; this results in feelings of anxiety and depression, as well as insomnia, stress and changes in the biological rhythm [25,26], which supports the results and alerts us to the possible medium and long-term repercussions of this reality.

Another dimension found to be significant included the factors related to the external locus of control (explaining about 7% of the relationship), with an emphasis on the relationship with others/social relationship and pro-social behavior (in SECs), and the relationship with the environment (in QoL). Relationship skills, one’s ability to initiate and maintain pro-social relationships, respect social norms and have good communication skills, seem to be a determinant in the perception of QoL, more specifically in individuals’ relationship with the environment (physical safety and security, home environment, financial resources, availability and quality of health and social care, opportunities for acquiring new information and skills, opportunities for recreation and leisure, physical environment and transport). As the Portuguese population has experienced moments of high social isolation [30,32], this seems to have been reflected in problems of loneliness and isolation [30,31], reduced physical activity, and an increased consumption of alcohol, cigarettes and unhealthy foods [33], which seems to explain the results of this relationship and entail repercussions that may extend over time, either socially, economically, behaviorally and/or psychologically [34].

### Limitations and Future Research

This study involved a whole set of social and psychological constructs that turned out to be important in order to understand the impact of the current context of war, pandemic and climate change on people’s lives. Although this study can be considered as preliminary research, as it needs to be further developed to fully understand how these changes impact the social and emotional competencies of individuals and their quality of life, the first results of this study show the importance of external/environmental factors on individuals’ QoL and how they affect individuals’ inner abilities, namely their capacity to make decisions. This study also shows that internal factors are those most important in explaining the relationship between SECs and QoL, with perceived psychological health (QoL) and self-management and motivation (SECs) being the variables that showed the greatest influence. The role played by sociodemographic variables (i.e., gender, age, educational level or occupation), both on SECs and QoL, however, remains to be understood.

The main limitations of the current study are related to the study design. Being a survey on the participants’ perception of social and emotional competencies and their quality of life, some individual biases may occur. Disseminating the survey via social networks, although it offered some advantages in terms of sample size, allowing a wide range of participants, had some limitations in terms of a controlled sample. The fact that this is a cross-sectional study does not allow a longitudinal perspective of the evolution of the relations studied, and the effect that the pandemic situation may have on this.

Taking into consideration the importance of the internal factors made known in this study and the results from the literature, for future research, it would be interesting to introduce a scale on empathy, in order to understand the relationship between this variable, QoL and SECs in the current context.

## 5. Conclusions

The current context of pandemics, war and climate change can be considered an atypical moment and a source of high levels of stress. Given this reality, the objective of this study was to assess, in a sample of Portuguese individuals, the relationship between the perception of social and emotional competencies and the perception of quality of life.

It was concluded that individuals have a lower perception of QoL in the environment domain, which seems to reflect the importance that external/environmental factors have on individuals and their QoL, generating the current context of insecurity, fear and instability when facing events beyond their direct control (external locus of control). The perception of SECs was also lower in the decision-making domain, which seems to represent a weakness in these individuals in terms of the reflexive consideration of different choices when taking into account the wellbeing of the self and of others; this may have medium and long-term high-risk repercussions at the level of societal dynamics.

The predictors of the relationship between SECs and QoL are mostly associated with internal factors, with perceived psychological health (QoL) and self-management and motivation (SECs) being the variables that showed the greatest influence. Secondly, there are external factors, with the relationship with others/social relationship and pro-social behavior (SECs), and the relationship with the environment (QoL), being the most important determinants.

According to the Portuguese Association of Psychologists [48], the cost of stress and psychological health problems at work have risen more than 60% in the last two years (5.3 billion), which validates the need to rethink health policies and direct professional practice in relation to these issues. Portugal is in the process of reformulating its mental health policies, and a National Coordination of Mental Health Policies of the Ministry of Health was established in December 2021 (DL 113/2021). These results may be extremely relevant in that reformulation and be an indicator for professional intervention/practice.

It is concluded that there is an urgent need to develop strategies for improving adaptive resilience, especially self-management and motivational skills, that will enable individuals to achieve a positive perception of psychological QoL and effectively overcome stressful life circumstances.

## Figures and Tables

**Table 1 behavsci-13-00249-t001:** Sociodemographic characteristics of the sample (N = 1139).

	n	%
Gender		
Male	303	26.7
Female	832	73.3
Age		
Adolescents (less than 18)	71	6.2
Young Adults (18 a 39)	552	48.5
Adults (40 to 59)	422	37.1
Older Adults (60 and more)	94	8.3
Nationality		
Portuguese	1070	93.9
Other	69	6.1
Marital status		
Single	576	50.6
Married / Living together	444	39.0
Separated/Divorced	107	9.4
Widow	12	1.1
Educational level		
Basic	144	12.6
Secondary	485	42.6
Third level	510	44.8
Occupation		
Student	345	30.3
Unemployed	48	4.2
Employed	691	60.7
Retired	54	4.7
Residence		
Small village	105	9.2
Large village	288	25.3
Town	744	65.4

Note: n—number of participants; %—percentage.

**Table 2 behavsci-13-00249-t002:** Descriptive values of the SEC_Q and WHOQOL_BREF.

	Min	Max	Mean	SD
SEC_Q				
Self-awareness	1.75	5.00	3.98	0.54
Self-management and motivation	1.00	5.00	3.85	0.68
Social awareness and pro-social behavior	1.00	5.00	3.97	0.50
Decision-making	1.00	5.00	3.83	0.76
Total	1.15	5.00	3.92	0.45
WHOQOL_BREF				
Physical	1.43	5.00	3.89	0.59
Psychological	1.00	5.00	3.70	0.64
Social relations	1.00	5.00	3.79	0.71
Environment	1.88	5.00	3.63	0.53
Total	2.04	5.00	3.75	0.48

Note: SD–standard deviation.

**Table 3 behavsci-13-00249-t003:** Correlations between dimensions of quality of life and social and emotional competences.

	Psy	Soc	Env	WTot	Saw	Smm	Sapb	Dm	SECTot
Phy	0.547	0.380	0.523	0.778	0.274	0.384	0.281	0.229	0.382
Psy		0.494	0.435	0.808	0.361	0.577	0.322	0.313	0.505
Soc			0.413	0.770	0.284	0.373	0.374	0.265	0.433
Env				0.737	0.314	0.325	0.351	0.250	0.411
Wtot					0.398	0.542	0.431	0.344	0.562
Saw						0.460	0.470	0.345	0.698
Smm							0.478	0.399	0.755
Sapb								0.422	0.836
Dm									0.722

Note: Phy—WHOQOL_Physical health; WHOQOL_Psy—Psychological health; Soc—WHOQOL Social relationships; Env—WHOQOL_Environmental health; Wtot—WHOQOL Total Score; Saw—SEC_Q Self-awareness; Smm—SEC_Q Self-management and motivation; Sapb—SEC_Q Social awareness and prosocial behavior; Dm—SEC_Q Decision-making. SEC_Tot—Social and emotional competences total. Note: All correlations significant at the 0.01 level.

**Table 4 behavsci-13-00249-t004:** Canonical Correlations, Eigenvalues and significance tests for each function.

Root	rc	rc2	Eigenvalue	Wilks	Num D.F.	Error D.F.	F	Sig.
1	0.619	0.383	0.620	0.572	16.000	3419.239	42.870	0.000
2	0.263	0.069	0.075	0.927	9.000	2725.936	9.588	0.000
3	0.063	0.004	0.004	0.996	4.000	2242.000	1.113	0.349
4	0.007	0.000	0.000	1.00	1.000	1122.000	0.059	0.807

Note: rc—canonical correlation; rc2—squared canonical correlation.

**Table 5 behavsci-13-00249-t005:** Canonical Solution for Emotional and Social Skills Predicting Quality of Life for Functions 1 and 2.

Variable	Function 1			Function 2			h2
	Coef	rs	rs2	Coef	rs	rs2	
Quality of Life							
Physical	−0.098	**−0.663**	0.440	−0.047	0.079	0.006	44.60
Psychological	−0.712	**−0.950**	0.903	−0.312	−0.300	0.090	**99.30**
Social relations	−0.224	**−0.684**	0.468	0.661	**0.502**	0.252	**72.00**
Environment	−0.169	**−0.623**	0.388	0.715	**0.568**	0.323	**71.10**
Emotional and Social Skills							
Self-awareness	−0.190	**−0.648**	0.420	0.154	0.270	0.073	49.28
Self-management and motivation	−0.738	**−0.952**	0.906	−0.308	−0.253	0.064	**97.03**
Social awareness and pro-social	−0.146	**−0.648**	0.420	0.602	**0.719**	0.517	**93.69**
Decision-making	−0.142	**−0.563**	0.317	0.064	0.212	0.045	36.19

Note: Coef = standardized canonical function coefficient; rs = structure coefficient; rs2 = squared structure coefficient; h2 = communality coefficient. Structure coefficients (rs) greater than |0.50| are bolded. Communality coefficients (h2) greater than 50 are bolded.

## Data Availability

Data may be made available upon request.

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
