# Peer review of "Association between Social and Emotional Competencies and Quality of Life in the Context of War, Pandemic and Climate Change"

_behavsci, 2023, doi:10.3390/bs13030249_

Round 1

Reviewer 1 Report

Dear Authors,

I would like to congratulate you for the relevance of the topic they have addressed and the rigour of the methodology you have applied in the study.

However, there are a number of aspects that need improvement, both in terms of form and content.

In terms of form, the structure follows only partially the criteria of a scientific article in that no space was reserved for the literature review (which should correspond to point 2 of the article, an point 3 to the methodology). 

Although this review was carried out, it is exclusively concentrated in the introduction. This aspect should be revised and more detail should be given regarding the sources consulted. In turn, the introduction should contain a final paragraph where the authors briefly describe the structure of the article, in relation to the topics addressed. 

In terms of language, the article is clear and easy to read. However, the verb tenses used in the abstract should be reviewed. Part of the sentences should be written in the present tense and not in the past tense. Besides, the abstract is incomplete, since there is no mention of the results obtained through the research developed. 

As far as materials and methods are concerned, a brief introduction to the type of research developed (quantitative) should be made, mentioning the sample selection criteria, before immediately beginning to describe it. This approach shall be carried out in the topic of the analysis of the results. 

In the conclusion, the limitations of the study should be presented, as well as proposals for future research. 

Finally, references 41 and 42 should be revised and adjusted to the citation rules designated by the Journal. 

These comments and suggestions should help you to improve the quality of your paper and its publication potential. 

Best regards,

The Reviewer

Author Response

Dear Reviewer,

We thank you for your time and attention in reviewing the article as well as your remarks and suggestions provided so that we could improve the quality of the paper and its publication potential. 

The response:

1.“I would like to congratulate you for the relevance of the topic they have addressed and the rigour of the methodology you have applied in the study.”

1.We thank you for the comment regarding the relevance of the topic and for the appreciation of rigor.

2.However, there are a number of aspects that need improvement, both in terms of form and content.

2.We attempted to improve the format and content according to the suggestions.

3.In terms of form, the structure follows only partially the criteria of a scientific article in that no space was reserved for the literature review (which should correspond to point 2 of the article, an point 3 to the methodology). Although this review was carried out, it is exclusively concentrated in the introduction.

3.As for the structure, we have effectively chosen to follow the directives of the Journal (Template), that is, in the Introduction, we took into consideration the directives “The current state of the research field should be carefully reviewed and key publications cited." and not that of a scientific article. Although we consider the indication relevant, we have chosen to maintain the journal's guidelines.

4.This aspect should be revised and more detail should be given regarding the sources consulted.

4.As for the consulted sources, we also chose to follow the directives of the Journal “References should be numbered in order of appearance and indicated by a numeral or numerals in square brackets—e.g., [1] or [2,3], or [4–6].”

5.In turn, the introduction should contain a final paragraph where the authors briefly describe the structure of the article, in relation to the topics addressed. 

5.A final paragraph has been introduced in the introduction with a brief description of the structure of the article (lines 122 and 123).

6.In terms of language, the article is clear and easy to read. However, the verb tenses used in the abstract should be reviewed. Part of the sentences should be written in the present tense and not in the past tense.

6.The verb tenses used in the abstract are reviewed and part of the sentences was written in the present tense.

7.Besides, the abstract is incomplete, since there is no mention of the results obtained through the research developed. 

7.The results were made more explicit (lines 26-27).

8.As far as materials and methods are concerned, a brief introduction to the type of research developed (quantitative) should be made, mentioning the sample selection criteria, before immediately beginning to describe it. This approach shall be carried out in the topic of the analysis of the results. 

8.The type of research developed was introduced (quantitative) (lines 122-123) and the sample selection criteria also (lines 128-131).

9.In the conclusion, the limitations of the study should be presented, as well as proposals for future research. 

9.Both limitations and proposals were introduced in 4.1 - Limitations and Future Research (lines 343-366).

10.Finally, references 41 and 42 should be revised and adjusted to the citation rules designated by the Journal. 

10.References 41 and 42 have been revised. 

Best regards,

Fátima Gameiro

Paula Ferreira

Miguel Faria

Reviewer 2 Report

I think this is potentially a publishable paper but I need clarification on the details in Table 5. I have worked frequently with factor analysis and I am puzzled by the mix of negative and positive correlations in your example of Function 2 (lines 257-263). 

In function 2, we can see that the criterion variables that contribute the most are social relations (.661) and environment (.715), (confirmed by the analysis of the structure coefficients), while in the predictor variable set the variables to consider is social awareness and pro-social behavior (1.02). Why is this correlation greater than 1, and why is self-management and motivation -.838 not mentioned with such a high correlation. 

 I understand the coefficients in Table 5 to be correlations. Please confirm that this understanding is correct. If I am incorrect, let me know. 

Author Response

Dear Reviewer,

We appreciate your time and attention in reviewing our article and the questions you pose.

The response:

1.I think this is potentially a publishable paper but I need clarification on the details in Table 5. I have worked frequently with factor analysis and I am puzzled by the mix of negative and positive correlations in your example of Function 2 (lines 257-263).  In function 2, we can see that the criterion variables that contribute the most are social relations (.661) and environment (.715), (confirmed by the analysis of the structure coefficients), while in the predictor variable set the variables to consider is social awareness and pro-social behavior (1.02). Why is this correlation greater than 1, and why is self-management and motivation -.838 not mentioned with such a high correlation. I understand the coefficients in Table 5 to be correlations. Please confirm that this understanding is correct. If I am incorrect, let me know. 

1. The coefficients in Table 5 are indeed correlations. All your doubts referred above are logical and understandable. What is harder for us to explain is the mistakes we have done when entering the numeric values, some manually, others through copy and paste. We have cleared them all and reentered them again. We checked the whole table and now we detected no mistakes. Since they are now correct, the text makes sense with the values in the table. Once again thank you for your attention.

Best regards,

Fátima Gameiro

Paula Ferreira

Miguel Faria

Reviewer 3 Report

The manuscript is well-written and structured and presents an interesting contribution to the journal. Only two minor suggestions are provided:

a) Please, further development of limitations and future research lines is needed.

b) A section about implications for policy and professional practice is highly recommended.

Author Response

Dear Reviewer,

We thank you for the care and the directions provided.

The response:

1.Please, further development of limitations and future research lines is needed.

1.Both limitations and proposals were introduced in 4.1 - Limitations and Future Research (lines 343-366).

2.A section about implications for policy and professional practice is highly recommended.

2.A paragraph was included about implications for policy and professional practice (lines 388-394).

Best regards,

Fátima Gameiro

Paula Ferreira

Miguel Faria

Round 2

Reviewer 2 Report

I am glad that the error in Table 5 has been corrected.

There are a number of small changes I recommend, some are necessary so that the article makes sense to native English speakers. I have noted these on the attached pdf.

Author Response

Dear Reviewer,

We thank you for your time and attention in reviewing the article as well as your remarks and suggestions provided so that we could improve the quality of the paper and its publication potential. 

The response:

1. I am glad that the error in Table 5 has been corrected.

1. Once again thank you for your attention.

2.There are a number of small changes I recommend, some are necessary so that the article makes sense to native English speakers.

2.. We are grateful for the suggestions that you have noted on the attached pdf.

2. We made the suggested changes in the lines 50, 109, 127-128, 134, Table 1 (Third level and occupation), lines 173, 183, 199 and 267.

Best regards,

Fátima Gameiro

Paula Ferreira

Miguel Faria
